# Maternal Mineral Nutrition Regulates Fetal Genomic Programming in Cattle: A Review

**DOI:** 10.3390/metabo13050593

**Published:** 2023-04-26

**Authors:** Muhammad Anas, Wellison J. S. Diniz, Ana Clara B. Menezes, Lawrence P. Reynolds, Joel S. Caton, Carl R. Dahlen, Alison K. Ward

**Affiliations:** 1Department of Animal Sciences, Center for Nutrition and Pregnancy, North Dakota State University, Fargo, ND 36849, USA; larry.reynolds@ndsu.edu (L.P.R.); joel.caton@ndsu.edu (J.S.C.); carl.dahlen@ndsu.edu (C.R.D.); 2Department of Animal Sciences, Auburn University, Auburn, AL 36849, USA; 3Department of Animal Science, South Dakota State University, Brookings, SD 57007, USA; anaclara.baiaomenezes@sdstate.edu; 4Department of Veterinary Biomedical Sciences, University of Saskatchewan, Saskatoon, SK S7N 5B4, Canada; alison.ward@usask.ca

**Keywords:** developmental biology, epigenetics, epigenome, essential nutrients, genetics, macrominerals, microminerals, restricted nutrition

## Abstract

Maternal mineral nutrition during the critical phases of fetal development may leave lifetime impacts on the productivity of an individual. Most research within the developmental origins of the health and disease (DOHaD) field is focused on the role of macronutrients in the genome function and programming of the developing fetus. On the other hand, there is a paucity of knowledge about the role of micronutrients and, specifically, minerals in regulating the epigenome of livestock species, especially cattle. Therefore, this review will address the effects of the maternal dietary mineral supply on the fetal developmental programming from the embryonic to the postnatal phases in cattle. To this end, we will draw a parallel between findings from our cattle model research with data from model animals, cell lines, and other livestock species. The coordinated role and function of different mineral elements in feto-maternal genomic regulation underlies the establishment of pregnancy and organogenesis and, ultimately, affects the development and functioning of metabolically important tissues, such as the fetal liver, skeletal muscle, and, importantly, the placenta. Through this review, we will delineate the key regulatory pathways involved in fetal programming based on the dietary maternal mineral supply and its crosstalk with epigenomic regulation in cattle.

## 1. Introduction

The mechanisms and the interaction of processes in the growth of an individual from embryonic to adult life is investigated in the field of developmental biology [1,2]. Rigorous research over the last half of a century has led to an emergence of reports that conjoin developmental biology with the areas of genetics [3], evolution [4], and epidemiology [5,6,7]. This impact on developmental biology, specifically in the area of epidemiology, has resulted in the formulation of a concept and, ultimately, a separate discipline, namely, fetal programing or the developmental origins of health and disease (DOHaD), first articulated by the human epidemiologist Dr. David Barker and colleagues [8]. 

The concept of the DOHaD hypothesis was not new at that time and can be traced back to the idea from the early 1800s of organic evolution, the concept that the adopted physical changes in one generation are transferable to subsequent generations via gametes, by Lamarck [9]; however, Barker was among the first to coin the term “fetal programming” or “developmental origins of health and disease”. Widdowson and McCance [10] were among the first cell biologists who provided evidence that there are some critical phases in development, especially during the pre-weaning period, in which undernutrition leads to changes in growth and development in rats. However, later evidence by Barker [8] and others in this emerging discipline suggested that fetal growth during pregnancy is regulated by numerous factors that, in turn, regulate genetic and epigenetic pathways [11,12]. Maternal nutrient intake during gestation is one of the major factors. The nutrients available to the conceptus not only affect the expression of the fetal and placental genomes but also significantly impact offspring growth postnatally [12,13]. 

Apart from human and laboratory animal model research [14,15], different studies have linked the concept of developmental programming to livestock performance, including cattle [16,17,18,19], sheep [20,21], and pigs [22,23]. Furthermore, some literature reviews [24,25,26,27,28] have pointed out the role of micronutrients in epigenome regulation, which leads to developmental programming. These ideas are the basis of developmental plasticity, defined as fetal adaptations to an altered intrauterine environment at the expense of postnatal developmental prospects [29]. Nutritional manipulations during times of developmental plasticity, i.e., embryonic, fetal, or neonatal life, exert either short or long-term effects on the development of muscle, adipose tissue, and ovarian reserves and the overall longevity of the offspring [13,30,31,32,33,34]. Based on these studies and some recent livestock modeling [35,36], it seems that nutritional alterations do affect metabolic disorders in humans or model organisms and may have equal or more severe impacts on cow–calf, feedlot, and dairy production systems in the livestock sector. 

The influence of macronutrients such as carbohydrates, proteins, and lipids has been researched and reviewed several times [17,25,37,38,39], with data extending to the omics level to investigate their regulatory roles in epigenomics and metabolomics [18,19,20,33,40]. In contrast, very little is known about their effects on the maternal dietary mineral supply and their regulatory role(s) in programming genomic function and fetal development. Although minerals are required in a smaller amount than macronutrients, their deficiency can lead to a significant reduction in growth and development [41]. Moreover, an excess of minerals can lead to toxicity. The mineral requirements of beef and dairy cattle along with the maximum tolerable limits are shown in Table 1 [42,43,44]. In this review, our objective is to summarize the key role of essential macro- and microminerals in fetal epigenome programming. 

## 2. Fetal Programming as a Multifactorial System

Although much of the research in fetal programming has focused on insults during mid- and late gestation, there is a growing interest in the earlier stages of gestation, which may “set the stage” for later programming events. Although most of the fetal size increase takes place in the last trimester, the first third of gestation is a critical period for organogenesis and tissue hyperplasia, as well as placentation [21,34,45,46], as shown in Figure 1. The prenatal growth trajectory of the conceptus is controlled by maternal nutrition either directly, by the provision of the essential nutrients; indirectly, via placental function, which regulates the transport of these nutrients [47]; or by altering the epigenetic mechanisms [48,49,50,51].

Pregnancy recognition occurs 15 to 16 days after estrous in cattle [58] and initiates maternal uterine vasculature changes to allow for the proper respiratory gas exchange and nutrient transfer to the developing embryo. Pregnancy is associated with a significant decrease in CpG methylation at the Sp1 promotor region of the *KCNMB1* gene (potassium calcium-activated channel subfamily M regulatory Beta subunit 1) and an increase in the expression of large-conductance Ca^+2^-activated K^+^ channel receptors, BK_Ca_, causing uterine artery dilation [59]. The expression of *KCNMB1* is also regulated by an increase in the expression of estrogenic receptor alpha [60] and the maintenance of membrane potential by K^+^ and Ca^+2^ ions to cause pregnancy-associated relaxation by reducing the myogenic tone [61,62] (Figure 2). In addition, estrogen is a potent angiogenic factor in the uterus and is involved in systemic cardiovascular changes during pregnancy (e.g., increased cardiac output [63,64]).

The development of fetal organs is significantly affected by mineral availability [65]. Zinc deficiency or excess during pregnancy can impact the development of multiple organs, including the brain, lungs, skeleton, and heart [66,67,68,69]. In the case of brain development, zinc deficiency impairs the function of *Zic* (Zinc finger proteins of the cerebellum) genes [70], which are essential for cerebellar development [70,71,72]. Zinc supranutrition (the supplementation of more than the normal requirement but less than the maximum tolerable limit) has shown the potential to enhance the cognitive ability of offspring in rats by increasing the expression of the signal transducer and activator of transcription 3 (*STAT3*) and matrix metalloproteinase-2/9 (*MMP-2/9*) [69]. *STAT3-MMP-2/9* activity promotes the invasion and migration of placental trophoblast cells and improves neural function [69,73]. In heart development, zinc deficiency alters the distribution of human natural killer-1 (*HNK-1*) cells and connexin 43 (*Cx43*) in the myocardium, contributing to the development of cardiovascular anomalies [74,75]. 

Like zinc, copper deficiency and supranutrition are associated with oxidative distress and neural degenerative disorders [76,77]. It has also been observed that both copper deficiency and the improper absorption of copper from the rumen in cattle can lead to such conditions. This can be due to the accessibility of molybdenum and sulfur, resulting in the formation of thiomolybdates, which can bind with copper in the rumen leading to inappropriate copper availability [78]. Copper deficiency leads to the suppression of the activity of a cuproenzyme, cytochrome-c oxidase, in the red nucleus region associated with large motor neurons, causing enzootic ataxia or swayback in lambs [79,80]. In addition, supranutritional levels of of copper, along with zinc, cobalt, and manganese, enhance the expression of metallothionein 1A (*MT1A*) in the dam and Cu-Zn superoxide dismutase (*CU/Zn SOD*) in developing offspring of cattle [77]. The MT1A is the major transporter of metal ions and *CU/Zn SOD* is involved in the regulation of oxidative stress and neurodegenerative disorder [77,81].

Selenium has biological functions via selenoproteins [28,82]. A maternal Se supplementation has effects on reproduction and developmental outcomes, which may be mediated by epigenetic events [28,83]. Selenium deficiency decreases the expression of selenoproteins, such as glutathione peroxidases, including *GPx1*, *GPx2*, *GPx4*, and *SELENOP*, including Selenoprotein-P, leading to alterations in embryonic development, oxidative stress mechanisms, reproductive development, and male fertility, respectively [84,85]. Maternal Se deficiency is associated with pancreatic atrophy in the developing fetus [86]. Moreover, maternal Se deficiency causes an elevation in the selenoenzyme type II deiodinase (*Dio2*), which reduces thyroxine production (Figure 3). On the other hand, supranutritional Se enhances thyroxine levels and impairs the growth hormone, insulin-like growth factor (GH-IGF) system [87,88]. The GH-IGF system impairment causes glucose intolerance and hypo-insulinemia in the fetus [86].

In cattle, primary and secondary myogenesis during the early fetal period start at days 47 and 119, respectively, and establish the lifetime potential for muscle development (see Figure 1) [56,93,94]. This is why there is no net increase in the number of muscle fibers after birth. Myogenesis requires the binding of active thyroid hormones to its receptors [95,96], especially thyroid receptor α, which is predominantly expressed in skeletal muscle [97] and is involved in promoting angiogenesis [98,99]. The deficiency of selenium, zinc, and iron can cause the impairment of thyroid hormone synthesis and action [100] by decreasing the expression of glutathione peroxidase (a selenoprotein) [101], 1,5′-deiodinase [102], and thyroperoxidase [103], respectively. Iron deficiency causes iron deficiency anemia, which reduces the activity of thyroid peroxidase (a Fe-dependent enzyme), leading to the repression of thyroid hormone synthesis and myogenesis [104]. 

Fetal skeletal development and bone mineralization are affected by maternal mineral status. Severe hypocalcemia, hypophosphatemia [105,106], and hypomagnesemia [107,108] are associated with reduced levels of parathyroid hormone (PTH) (Figure 4). Normally, *PTH* acts on PTH receptor 1 (*PTHR1*) of the kidney, which activates cyclic adenosine monophosphate (*cAMP*)-associated pathways and requires Mg^+2^ as a cofactor [109,110] for calcitriol (1,25(OH)D) and vitamin D production. However, a decrease in PTH leads to vitamin D deficiency [111], which ultimately affects bone formation and mineralization in the developing fetus [107]. Both hypo- and hypercalcemic dams were associated with the upregulation of fatty acid binding protein 4 (*FABP4*), fatty acid synthase (*FASN*), and acetyl coenzyme A carboxylase 1 (*ACC1*) in the adipose tissue and liver of the offspring, leading to dyslipidemia and bone demineralization to maintain the serum level of calcium in the offspring [112].

## 3. The Roles of Minerals in Fetal Genome Regulation

Epigenomic regulation in the fetus is affected by the insufficiency of maternal nutrients, including minerals. In a recent study in cattle, the authors reported that essential micronutrient supplementation and the dietary plane of nutrition (as assessed by the rate of bodyweight gain) during early gestation can affect the neonatal immune response and the availability of mineral reserves for postnatal development [114,115]. In addition, the expression of genes associated with cholesterol synthesis, ion homeostasis, and nutrient transport was altered in the developing placenta [116,117]. To explain the epigenomic regulation of these pathways based on maternal mineral homeostasis and their effects on fetal programming, different studies have been conducted in model organisms (see Table 2).

The transcriptomics of the developing fetus can be regulated either by changing the methylation pattern of specific DNA regions or by repressing mRNA expression based on the adequacy of maternal mineral nutrition. For example, Zn is transported in the blood by forming a complex with metallothionein-2 (*MT2)* [126]. In a mouse model, Zn deficiency was associated with a polymorphism in *MT2* at rs1610216 (MT2A–209A/G) along with histone modification and hypermethylation of a metal-responsive element (MRE) in the promotor region of *MT2* [123], and similar changes are seen in human [127]. Furthermore, the mRNA expression of zinc transporter 1 (*ZnT1*) in a zebrafish model [128], as well as zinc importing protein (*ZIP14*) in a rat model [75], and divalent metal transporter 1 (*DMT1*) in human cell lines [129] is associated with the availability of zinc. The impairment of the expression of these genes affects zinc availability to the developing fetus and, subsequently, organogenesis [66,67].

Selenium deficiency affects the expression of selenoenzymes such as *GPx1*, *GPx2*, and *GPx4* and selenoproteins (i.e., Selenoproteins-P, *SELENOP)*, which are involved in not only fetal reproductive development and the regulation of oxidative stress [84,85] but also the overall DNA methylation pattern. Selenium availability affects the concentration of S-adenosyl homocysteine (a potential inhibitor of DNA methyltransferases) and the availability of S-adenosyl methionine (the methyl-donor for all methylation reactions) in the methionine–homocysteine cycle [130,131,132,133]. In a rainbow trout model, selenium availability affected the differentially methylated cytosines of more than 6500 differentially methylated genes associated with immune modulations and neural signaling [124].

Like zinc and selenium deficiency, maternal iron deficiency in pregnancy is critical as it can permanently affect brain development [120,121]. Iron deficiency is associated with histone modification and DNA methylation at the brain-derived neurotrophic factor IV (*BDNF-IV*) promotor region in the hippocampus of the developing fetus, which affects cognitive response and hippocampal plasticity, as observed in a rat model [120]. Additionally, iron deficiency causes a reduction in the expression of DMT1 (a major transporter) in rats, which leads to impaired manganese availability to the developing fetus [134,135]. Copper deficiency can also impair *DMT1* expression and affect the availability of iron and manganese to the fetus [136] (Figure 5). In a study of copper availability and DNA methylation changes in fetuses during pregnancy in humans, the most robust negatively associated, differentially methylated region was found in a zinc-finger gene, *ZNF197*, which was correlated with birth weight [125]. 

In terms of macrominerals, maternal magnesium deficiency and calcium deficiency affect the methylation of CpG island regions of hydroxysteroid 11-beta dehydrogenase 2 (*Hsd11b2*) [119] and hydroxysteroid 11-beta dehydrogenase 1 (*Hsd11b1*) [118], respectively, leading to the impairment of glucocorticoid metabolism in the developing fetus. The hepatic glucocorticoid concentration was altered in rats [119], which affects the GH-IGF system, leading to a reduction in postnatal skeletal development and myogenesis [95,96]. Due to sex-specific modifications, however, the effects of insulin resistance were minimal in later F2 and F3 generations, although insulin production was still dysregulated [140,141]. 

## 4. Feto-Maternal Crosstalk

After attachment/implantation, fetomaternal crosstalk and the transport of nutrients (including minerals) are completely dependent on the placenta. Before placental vascularization is completed (the first 50 d of gestation in cattle), histotrophic nutrition (via uterine secretions) is the main source of nutrients to the fetus [21]. However, after placental circulation is established, hemotrophic nutrition is the primary pathway involved in the transfer of nutrients to the fetus. In ruminants, chorionic development begins at about day 20 of pregnancy, and placental development along with the interdigitation of fetomaternal villi completes at day 50 [46]. The establishment of pregnancy requires minerals and is especially associated with membrane potential as regulated by estrogen, K^+^, and Ca^+2^ [61,62]. In ewes, Ca^+2^ and Na^+^ levels increase in histotroph on post-fertilization from days 13 to 16, suggesting their role in placental development and implantation [142]. During pregnancy, estrogen binds to the SP1 site in the promotor of the *KCNMB1* gene, which causes upregulation in the expression of BK_Ca_ [61]. The opening of BK_Ca_ channels results in the efflux of K^+^ and the sarcoplasmic release of Ca^+2^ to further enhance *KCNMB1* expression and BK_Ca_ channels [59], which causes the pregnancy-induced relaxation of uterine vascular smooth muscle. This hypothesis was further strengthened in cattle models by one recent study in which sarcoplasmic reticulum Ca^+2^-ATPase 3 (*ATP2A3*) and ATPase subunit beta-1 (*ATP1B1*) were found to be upregulated in caruncles of the mineral-supplemented group compared to the non-supplemented one, indicating the intracellular sarcoplasmic Ca^+2^ release through these ATPase pumps [116]. 

The placental transport of other micro- and macrominerals is required for fetal developmental programming. Zinc is maternally transported by *ZnT1* in the form of the Zn-MT complex [126] and binds to *ZIP14* in placental trophoblast. It is further transported to the fetus by *ZnT2* and *DMT1* [126] (Figure 5). When zinc and copper supplementation and cobalt and manganese in beef cattle were examined together, *MT1A* expression was found upregulated in the dams and Cu/Zn SOD levels were upregulated in their successive offerings [77]. This explains the maternal pathway for zinc transport, but the transport mechanism of zinc in the fetus still needs validation in the cattle model. The cattle model study, however, recently showed evidence supporting fetal transport. It was found that the metallothionein coding genes *MT1A*, *MT2A*, and *MT1E* are upregulated, while *ZnT10* is differentially expressed in mineral-supplemented fetal groups compared to non-supplemented groups [117].

Selenium and iodine deficiency can impair fetal development by influencing the GH-IGF system [87,91]. In the maternal liver, selenium in the presence of selenophosphate synthetase and *SEPSECS* (Sep [O-Phosphoserine]) TRNA:Sec ([Selenocysteine TRNA Synthase]) is converted into selenophosphate and selenocyctenyl tRNA, respectively. Selenocyctenyl tRNA in the presence of *SBP* (selenocysteine binding protein 2) produces selenoproteins, such as *SELENOP*, and deiodinases [89]. *SELENOP* binds to *ApoER2* (Apolipoprotein E Receptor-2) in the placental trophoblast and is transported to the fetus [85,90]. The production of *Dio2* converts the Thyroxine-4 (*T4*) to active triiodothyronine (*T3*) [89]. T3 and T4 are produced in the fetal thyroid gland by thyroglobulin produced from iodine and tyrosine [91]. Free T3 increases the expression of *Dio3* in placental trophoblast, which will convert T3 back to inactive T4 in the fetus. This mechanism is associated with low active T3 in the fetus and, thus, prevents fetal hyperthyroidism [92] (Figure 3). The proposed mechanism of selenium feto-maternal transport based on lab animal data was not validated in a lamb model, and no changes in fetal T3:T4 were identified when ewes were supplemented with selenium [88]. This puts a question mark on the role of *Dio3* in the interconversion of T3 to T4 in fetuses, indicating the current lack of understanding regarding fetal selenium transport. 

Placental transporters are regulated by minerals, including manganese, iron, and copper, as shown in Figure 5. Manganese or iron in blood serum forms a complex with transferrin (*Tf*) proteins. This complex interacts with the transferrin 1 receptor (*TfR1*) at the placental microvillous membrane and is endocytosed in vesicle form [135,137]. This proposed pathway of the manganese/iron-*Tf* complex binding to *TfR1* for vesicular endocytosis was also supported in a recent cattle model study, in which *TfR1* was differentially expressed in the minerals-supplemented group compared to the non-supplemented group [117]. The acidification of these vesicles causes the release of either manganese or iron, which is further transported to the fetus by *DMT1* [135,137,139,143,144]. The fetal uptake of Cu is performed by the binding of Cu^+2^ from maternal plasma to a high-affinity copper transporter protein 1, *CTR1*, at the placental trophoblast [145]. The transfer of Cu from *CTR1* to a chaperon protein, *ATOX1*, is associated with the transport to either the fetus by the ATPase *ATP7A* or back to the dam and then the maternal liver by the ATPase *ATP7B* [138,139]. Copper supplementation data in a cattle model have shown that, rather than *CTR1*, *MTIA* and *Cu/Zn SOD* can be the maternal and fetal transporters of copper, respectively [77,146], emphasizing the need for further research on copper deficiency in a cattle model.

The transport and regulation of some macrominerals such as magnesium, calcium, and phosphorous are interlinked and are affected by the availability of each in the maternal diet (Figure 4). Calcium from blood plasma interacts with the calcium-sensing receptor (*CaSR*) at the placenta and increases the expression of placental parathyroid hormone-related proteins (*PTHrP*) and the release of parathyroid hormone (*PTH*) from the fetal parathyroid gland [105,113]. *PTHrP* expression increases in the maternal mammary tissues of humans; thus, both calcium-associated mechanisms run parallel in the fetal and maternal systems [111]. Both *PTH* and *PTHrP* from the fetus and *PTHrP* from the dam act on PTH receptor 1 (*PTHR1*) of the kidney both in the fetus and dam, which activates cyclic adenosine monophosphate (*cAMP*)-associated pathways and requires Mg^+2^ as a cofactor [109,110]. The *cAMP*-associated pathways increase the expression of 25(OH)D(3)-1-α hydroxylase, which then increases calcitriol production from 25(OH)D3, resulting in calcium and phosphorous absorption from the intestine [110,111]. Based on these mechanisms, the deficiency of any of these minerals—magnesium, calcium, or phosphorous—will affect the availability of each other and, ultimately, affect bone mineralization and development in the fetal and maternal systems.

## 5. Final Considerations

The potential regulatory roles of maternal mineral intake in developmental programming, from conception to birth, indicate that the excess or deficiency of minerals can lead to pre- and postnatal metabolic disorders and growth abnormalities [147]. The limitation of the studies we have cited is that most are based primarily on laboratory animal models. Although there is a need to validate these observations more extensively and in other species, basic epigenomic regulation involves similar patterns in most species, and, as such, similar responses are expected in cattle and other livestock models [148,149,150]. Recent studies using cattle as experimental models [18,19,114,115,116] have provided evidence that maternal nutrition during early pregnancy affects the deposition of minerals for postnatal development, metal ion homeostasis, growth regulatory pathways (e.g., the GH-IGF and thyroid hormone pathways), and, in particular, the overall metabolomics of the developing fetus. Based on animal models, and despite the limited evidence from livestock models [116,117], we believe that the proposed mechanisms are very likely to be valid but still need to be examined in livestock models of mineral deficiency or excess. Moreover, the literature concerning sheep [88] and cattle [77,146] models still brings into question the accepted pathways of feto-maternal transport, especially for selenium, zinc, and copper.

Another major limitation is that the available data are mostly based on mRNA, which is insufficient to make conclusions about epigenomic regulation and its role in fetal programming. In this review, we have presented some key genomic/molecular regulatory pathways involved in the effects of minerals on the epigenetic regulation of fetal and placental development. However, much more information and a much better understanding of the alterations in gene expression and their association with epigenetic signals (i.e., non-coding RNAs, histone modifications, and DNA methylation patterns) are needed in other species models. Recent reviews have done a good job explaining the role of epigenetic signals in the effects of maternal and paternal stressors on the developing embryo and fetus [34,151]. The effects of these stressors, however, across the offspring’s lifetime and across subsequent generations warrant further investigation. In addition, many of the mineral deficiencies regulate the GH-IGF and thyroid hormone systems; however, the compensatory adaptations of subsequent generations to maternal and postnatal stressors in terms of epigenetic signal inheritance need to be explored further [140]. 

Future studies are needed to address these abovementioned limitations in the available data to gain a better understanding of maternal mineral nutrition and its role in the epigenetic regulation of the developing fetus and offspring. Furthermore, we also need to identify efficient ways to correlate the epigenomic signaling data (i.e., the DNA methylation pattern, histone modifications, or non-coding RNAs) with genomic regulation, so that we can effectively translate the effects of maternal nutrition and other prenatal stressors across generations by delineating the cascade of minerals and mineral transporters involved in feto-maternal crosstalk. However, based on the available literature from animal models (including livestock species), humans, and cell lines, we have discussed our current understanding of the genomic regulatory roles of the following:Calcium in dyslipidemia and insulin resistance;Zinc in neural, cardiac, and general organ development and trace mineral transport;Selenium in reproductive function, the regulation of the GH-IGF system, and the thyroid hormone system;Magnesium in glucocorticoid metabolism;Copper in oxidative stress, the regulation of the GH-IGF system, and placental development;Calcium and potassium in the establishment of pregnancy and the regulation of placental vascular tone;Selenium and iron in growth hormone metabolism and myogenesis;Magnesium, calcium, and phosphorous in skeletal development and parathyroid hormone and vitamin D metabolism.

## Figures and Tables

**Figure 1 metabolites-13-00593-f001:**
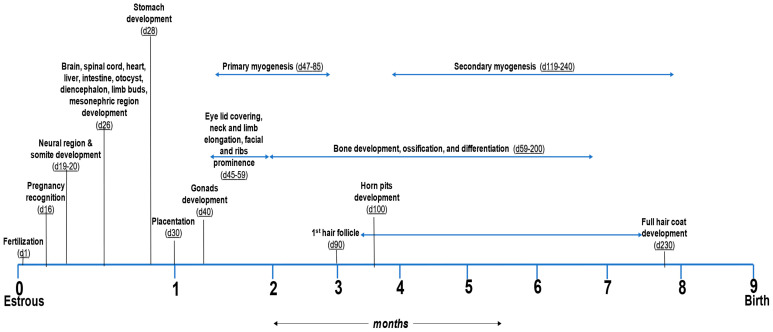
Timeline indicating organogenesis and development of different structures during gestation in cattle [45,46,52,53,54,55,56,57,58].

**Figure 2 metabolites-13-00593-f002:**
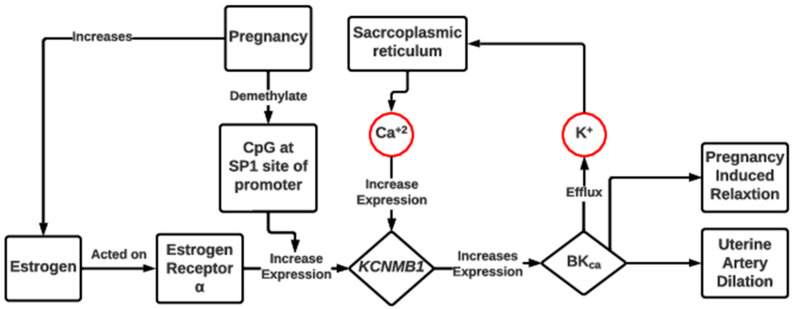
Schematic overview of the role of estrogen, Ca^+2^, and K^+^ in the establishment of pregnancy-associated relaxation and myogenic tone reduction. Pregnancy increases the basal estrogen level and demethylate CpG at the Sp1 promotor site to increase *KCNMB1* expression [61]. *KCNMB1* leads to opening of *BKca* channels, causing efflux of K^+^ and release of Ca^+2^, which further promote *KCNMB1* expression [59]. Change in membrane potential of vascular smooth muscle in the uterine artery by efflux of K^+^ and release of Ca^+2^ results in uterine artery dilation along with pregnancy-induced relaxation [61]. *KCNMB1,* potassium calcium-activated channel subfamily M regulatory Beta subunit 1; BK_Ca_, large-conductance Ca^+2^-activated K^+^ channel receptors.

**Figure 3 metabolites-13-00593-f003:**
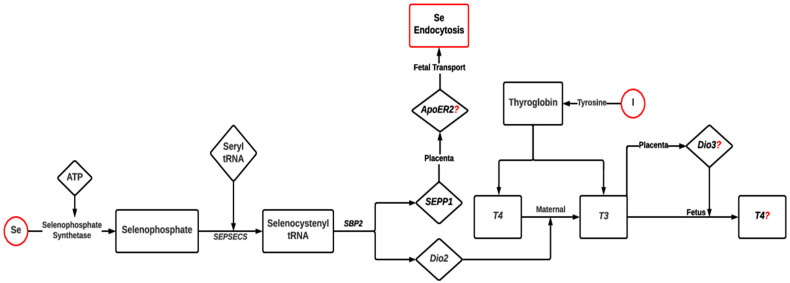
Proposed mechanism of fetal–maternal transport of selenium and iodine and their role in thyroxine metabolism in cattle. Maternal selenium concentration impacts the expression of selenoproteins, i.e., *SEPP1* and *Dio2* in dam’s liver [89]. *SEPP1* will be transported to fetus by *ApoER2,* and *Dio2* will affect thyroxines interconversion across fetal–maternal tissues [85,89,90,91,92]. *SEPSECS*, (Sep (O-Phosphoserine) TRNA:Sec (Selenocysteine) TRNA Synthase); *SBP2*, selenocysteine binding protein 2; *SEPP1*, Selenoprotein-P; *Dio2*, TypeII Deiodinase; *ApoER2*, Apolipoprotein E Receptor-2; *Dio3*, TypeIII Deiodinase; *T3*, Triiodothyronine-3; *T4*, Thyroxine-4.

**Figure 4 metabolites-13-00593-f004:**
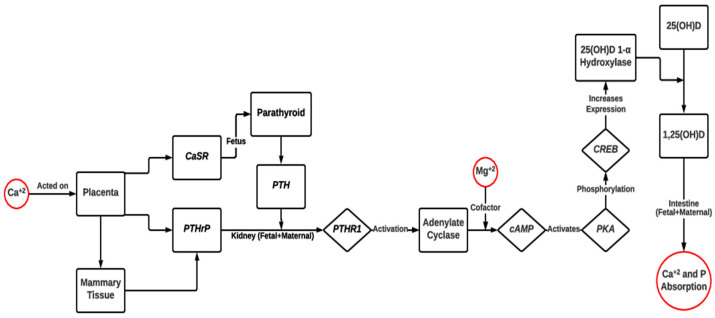
Proposed mechanism of feto-maternal transport of Ca, P, Mg, and iodine and their roles in regulating molecular mechanism of parathyroid hormone (*PTH*) and calcitriol (1,25-dihydroxycholecalciferol, 1,25(OH)2D) in cattle. Maternal Ca^+2^ concentration causes a change in the expression of *PTHrP* in placenta and mammary tissue along with the expression of *CaSR* in placenta [105,111,113]. *PTH* is regulated by *CaSR* in the fetus and maternal *PTHrP* binds to *PTHR1* in the kidney and activates the cAMP-associated conversion of calcitriol (1,25(OH)D), in which Mg^+2^ will be used as a cofactor [109,110,111]. *CaSR*, calcium sensing receptor; *PTHrP*, parathyroid hormone related proteins; *PTHR1,* parathyroid hormone 1 receptor; *cAMP*, cyclic adenosine monophosphate; *PKA*, phosphokinase activated; *CREB*, cAMP response element-binding protein; calcitriol, 1,25(OH)D.

**Figure 5 metabolites-13-00593-f005:**
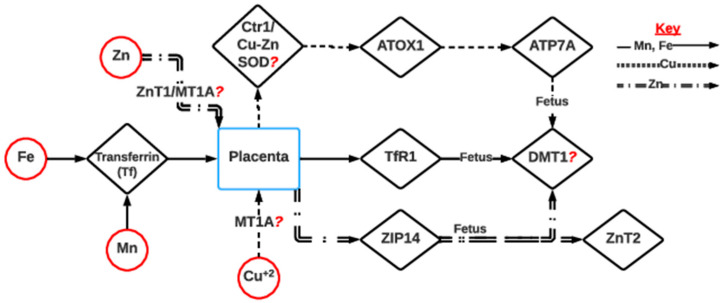
Proposed mechanism of fetomaternal transport of Fe, Mn, Zn, and Cu in cattle. Iron or manganese can form complexes with transferrin and bind to *TfR1* in the placenta, which transports Fe or Mn to the fetus [135,137]. Cu and Zn are also transported by placental transporters *CTR1* or *Cu-Zn SOD* [77,81,138,139] and *ZIP14* [75], respectively. All these mentioned minerals are in the divalent form and, in the fetus, are transported via *DMT1* [129,134,135,136]. *ZnT1*, zinc transporter 1; *MT1A*, metallothionine-1A; *ZIP14,* zinc-importing protein; *DMT1,* divalent metal transporter 1; *CTR1,* copper transporter protein 1; *Cu-Zn SOD*, copper zinc superoxide dismutase; *ATOX1*, antioxidant 1 copper chaperone; *ATP7A*, ATPase copper-transporting alpha; *TfR1*, transferrin 1 receptor.

**Table 1 metabolites-13-00593-t001:** Mineral requirements and maximum tolerable limits during pregnancy as established by the National Academies of Sciences, Engineering, and Medicine (NASEM) for cattle.

	Mineral Requirements of Dairy Cattle ^a^			Mineral Requirements of Beef Cattle ^b^			Maximum Tolerable Level (MTL) ^c^
Mineral	Lactating Cow	Dry Pregnant Cow	Growing Heifer	Growing and Finishing Cow	Gestating Cow	Early Lactating Cow	
Calcium, % ^d^	0.59	0.35	0.45	0.6	0.25	0.3	1.5
Phosphorous, %	0.36	0.2	0.21	0.22	0.17	0.21	0.7
Magnesium, %	0.17	0.14	0.12	0.1	0.12	0.20	0.40
Potassium, %	1.02	0.66	0.56	0.6	0.6	0.7	2
Sodium, %	0.22	0.17	0.16	0.06–0.08	0.06–0.08	0.1	3
Sulfur, %	0.2	0.2	0.2	0.15	0.15	0.15	0.4
Cobalt, mg/kg ^d^	0.2	0.2	0.2	0.15	0.15	0.15	25
Copper, mg/kg	9	18.5	15.75	10	10	10	40
Iodine, mg/kg	0.44	0.53	0.55	0.5	0.5	0.5	50
Iron, mg/kg	17.6	14	32.5	50	50	50	500
Manganese, mg/kg	28	40.5	41.25	20	40	40	2000
Selenium, mg/kg	0.3	0.3	0.3	0.1	0.1	0.1	5
Zinc, mg/kg	60.8	31	36.5	30	30	30	500

^a^ Nutrient Requirements of Dairy Cattle from the NASEM, 2021 [42]; ^b^ Nutrient Requirements of Beef Cattle by the NASEM, 2016 [43]; ^c^ Mineral maximum tolerable levels (MTL) established for cattle by the NASEM, 2005 [44]; ^d^ % of dry matter and mg/kg of dry matter.

**Table 2 metabolites-13-00593-t002:** Maternal mineral nutrition associated with epigenomic regulation of the developing fetus.

Mineral	Model	Epigenome Regulation	Organ	Effect	Reference
Ca	Rat	Hypomethylation of CpG dinucleotide in promotor of hydroxysteroid 11-beta dehydrogenase 1 *(Hsd11b1)*	Liver	Induction of insulin resistance in adult life	[118]
Mg	Rat (Mg deficient model)	Hypermethylation of CpG dinucleotide in promotor of 11β-hydroxysteroid dehydrogenase-2 (*Hsd11b2*)	Liver	Alters neonatal hepatic glucocorticoid metabolism	[119]
Fe	Rat (Fe deficient model)	Hypomethylation at CpG site and reduction in histone H4 acetylation in promoter of brain-derived neurotrophic factor (*BDNF*)	Brain (hippocampus)	Crucial for regulation of hippocampal plasticity and development of neural circuit	[120]
Fe	Rat (Fe deficient model)	Hypermethylation in 63 genes and hypomethylation in 45 genes	Brain (hippocampus)	Neural function dysregulation and alterations in cell-to-cell signal transduction	[121]
Fe	Domestic pig (Fe deficient model)	Twelve differentially methylated cytosines regulating nine differentially expressed genes were identified	Brain (hippocampus)	Associated genes were found to be involved in angiogenesis and neurodevelopment	[122]
Zn	Mouse (Zn deficient model)	Elevated expression of metallothionine-2 (*MT2*) mRNA response to histone modifications in metal-responsive elements associated with the promotor region of *MT2*	Liver	Epigenetic memory of zinc deficiency in early development may persist to adulthood, impacting availability of essential trace minerals	[123]
Se	Rainbow trout (Se deficient model)	Selenium availability affected the differentially methylated cytosines of more than 6500 differentially methylated genes	Liver	The 6500 differentially methylated genes were found to be associated with immune modulations and neural signaling	[124]
Cu	Humans	Copper levels positively coincided with DNA methylation at CpG island and transcription site of Zinc Finger Protein 197 (*ZNF197*)	Placenta	Can alter placentation and growth in postnatal life by impairing growth hormone secretion	[125]

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
