# Peer review of "Maternal Mineral Nutrition Regulates Fetal Genomic Programming in Cattle: A Review"

_metabolites, 2023, doi:10.3390/metabo13050593_

Round 1

Reviewer 1 Report

This manuscript is a review of the role of minerals in fetal programming and epigenetics. This is a thorough review of the molecular basis of minerals in fetal development and epigenetic effects.

My largest concern of this review is the lack of post-parturient data presented. Studies presented seem to focus on changes in early gestation, which is fine, but the studies were terminated during gestation to collect fetal tissue samples. The big question left unanswered is whether the molecular changes occurring in early gestation are large enough to cause differences in economically-relevant traits in the offspring post-calving. The authors should include data from studies of this type or include a discussion of the lack of these types of studies.

Specific comments:

L24 - 'correspond some' is awkward wording

L62 - the 'do not affect' does not seem to fit with the beginning of the sentence or the theme of the paragraph

L173 - I am not sure 'alleviated' is the correct word here

L273-275 - this sentence is difficult to understand. The comma after 'zinc' indicates that there should be some focus on what happens when examining zinc, but the rest of the sentence does not complete that thought. Also, 'while Cu/Zn SOD levels' implies that you should tell me what happens to these levels, but that is missing.

L304 - change to '[139,140]. Copper supplementation'

L306 - change to 'respectively [78,147],'

L316 - is this the PTH receptor 1 in the fetal or maternal kidney? need to make sure whether you are talking about fetal or maternal tissues is clear throughout the manuscript.

308-323 - How is fetal PTH involved because there is no calcium in the fetal GI tract? Does fetal PTH cross to maternal blood to stimulate cAMP pathways?

L336 - change to 'lab animal models'. Also, should search entire manuscript to designated what type of animal models, because the term 'animal models' is used to refer to both lab or livestock animals.

L356 - spell out those limitations

L494 - change to 'differentiation'

Figures - all figures are somewhat blurry

Reviewer 2 Report

The manuscript presented for the review entitled "Maternal Mineral Nutrition Regulates Fetal Genomic Programming in Cattle: A Review" undertakes an important and interesting issue.

I have however some comments which are as follows:

The autors state in the Abstract: "....we will draw a parallel between findings from our cattle model research with data from model animals, cell lines, and other livestock species." - I can't see authors findings in the paper, they should be emphasized in some way as declared in the abstract

Figure 1 - what do the letters on the horizontal axis mean? I suppose that these ate months, but this should be described

Figures 3 and 5 - what do the question marks mean? This should be explained clearly.

line 161 - remove "in cattle"

Paragraph 2 - the content does not reflect the title; information about particular minerals should be moved rather to paragraph 3

Reviewer 3 Report

Dear Authors,

the present manuscript (metabolites-2356958 "Maternal Mineral Nutrition Regulates Fetal Genomic Programming in Cattle: A Review") is important and actual work. It is in a frame of the major journal scopes. This review addresses the role of “maternal dietary mineral supply on the fetal developmental programming from embryonic to the postnatal time phases in cattle”. The authors compare their research of “cattle model with data from model animals, cell lines, and other livestock species”. It is important to highlight the use of some micro- and macroelements (MMEs) “in fetal-maternal genomic regulation underlies the establishment of pregnancy and organogenesis, and ultimately affects the development and function of metabolically important tissues such as the fetal liver and skeletal muscle and, importantly, the placenta”. The authors present some of the “key regulatory pathways involved in fetal programming based on dietary maternal mineral supply and its crosstalk with epigenomic regulation in cattle”.  

It can be useful, if the authors can provide small corrections in the text. In particular, 

1. To add in the “Abstract” (after small text reduction) some very important statements from the part 5 “Final Considerations” (from the Lines 364 to 376) as the following: “…..we have discussed our current understanding of the genomic regulatory roles of: • calcium in dyslipidemia and insulin resistance; • zinc in neural, cardiac, general organ development, and trace mineral transport; • selenium in reproductive function, regulation of the GH-IGF system, and the thyroid hormone system; • magnesium in glucocorticoid metabolism; • copper in oxidative stress, regulation of the GH-IGF system, and placental development; • calcium and potassium in the establishment of pregnancy and the regulation of placental vascular tone; • selenium and iron in growth hormone metabolism and myogenesis; and • magnesium, calcium, and phosphorous in skeletal development along with parathyroid hormone and vitamin D metabolism”.

2. To make some text reduction in the part 1 “Introduction” (i.e. historical overview from the Lines 39 to 51) starting from “The concept of the DOHaD hypothesis was not new at that time and can be traced back to the early 1800s….”, but keep the valuable references ([9-13]).

3. To add some data on Se from the text (Lines 215-223 and, may be, 224-228) also to the Table 2: “Maternal mineral nutrition associated with epigenomic regulation of the developing fetus” (Line  202).

Reconsider after minor revision. 
